# Identification of Key Residues Required for RNA Silencing Suppressor Activity of p23 Protein from a Mild Strain of Citrus Tristeza Virus

**DOI:** 10.3390/v11090782

**Published:** 2019-08-25

**Authors:** Zhuoran Li, Yizhong He, Tao Luo, Xi Zhang, Haoliang Wan, Atta Ur Rehman, Xinru Bao, Qian Zhang, Jia Chen, Rangwei Xu, Yingtian Deng, Yunliu Zeng, Juan Xu, Ni Hong, Feng Li, Yunjiang Cheng

**Affiliations:** 1National R&D Center for Citrus Preservation, Huazhong Agricultural University, Wuhan 430070, China; 2Key Laboratory of Horticultural Plant Biology, Ministry of Education, Huazhong Agricultural University, Wuhan 430070, China; 3Key Lab of Plant Pathology of Hubei Province, College of Plant Science and Technology, Huazhong Agricultural University, Wuhan 430070, China

**Keywords:** citrus tristeza virus, mild strains, RNA silencing suppressor activity, p23 point mutation

## Abstract

The severe strain of citrus tristeza virus (CTV) causes quick decline of citrus trees. However, the CTV mild strain causes no symptoms and commonly presents in citrus trees. Viral suppressor of RNA silencing (VSR) plays an important role in the successful invasion of viruses into plants. For CTV, VSR has mostly been studied in severe strains. In this study, the N4 mild strain in China was sequenced and found to have high sequence identity with the T30 strain. Furthermore, we verified the functions of three VSRs in the N4 strain, and p23 was found to be the most effective in terms of local silencing suppressor activity among the three CTV VSRs and localized to both nucleus and plasmodesmata, which is similar to CTV T36 strain. Several conserved amino acids were identified in p23. Mutation of E95A/V96A and M99A/L100AA impaired p23 protein stability. Consequently, these two mutants lost most of its suppressor activity and their protein levels could not be rescued by co-expressing p19. Q93A and R143A/E144A abolished p23 suppressor activity only and their protein levels increased to wild type level when co-expressed with p19. This work may facilitate a better understanding of the pathogenic mechanism of CTV mild strains.

## 1. Introduction

Citrus tristeza virus (CTV) is one of the most devastating viruses causing severe losses in the citrus industry all over the world [1]. CTV symptoms observed on citrus plants mainly include yellowing of seedlings, rapid decline of trees, and stem pitting. Moreover, CTV strains were found to have a high level of sequence diversity, which may be due to three possible reasons: (1) diversification of major ancestral CTV lineage; (2) conservation and co-evolution of major functional domains within CTV genotypes; and (3) extensive recombination between lineages [2]. To date, more than 100 strains of CTV have been identified and complete genome sequences are available for 51 of them. Phylogenetic analyses have divided these 51 strains into six groups: T36, T3, T30, VT, B165, and RB. T36 and T3 strains, found in Florida, can induce rapid decline of citrus trees [3,4]. The T30 strain, whose infection does not cause obvious symptoms in citrus, was isolated from *Citrus sinensis* cv. Hamlin in Florida and sequenced in 2000 [5]. VT, which causes rapid decline and stem pitting in grapefruit, was isolated from the Israeli sweet orange and sequenced in 2004 [6,7]. B165, a CTV isolate from *Citrus reticulata* cv. Ellendale, was sequenced in India in 2009 and found to cause severe decline in Mexican lime, yellowing in Duncan grapefruit, and yellowing and stem pitting symptoms in lime [8]. RB strain was detected from *Poncirus trifoliata* and sequenced in New Zealand in 2005, and was found to break the resistance of trifoliate against CTV [9]. A high level of CTV genetic diversity brings challenges to the management of this viral disease in citrus. 

RNA silencing is a natural antiviral defense mechanism operating in both plants and animals [10]. Replicating viral double-stranded (ds)RNA intermediates and dsRNA made from single-stranded viral RNA by plant RNA-dependent RNA polymerase (RDR) are diced into 21- to 24-nucleotide (nt) small interfering (si)RNAs by plant Dicer-like enzymes. The virus-specific siRNAs are incorporated into plant Argonaute protein to form RNA-induced silencing complex (RISC). RISC binds and slices viral mRNAs guided by its associated virus-derived siRNA, thus restricting viral gene expression and replication in host cells. In plants, viral siRNA moves from cell to cell and systemically restricts virus spreading. SiRNA-based transgenic approaches have been applied to engineer a variety of crops expressing artificial small interfering RNAs or microRNAs directly targeting viral genes for viral resistance, including CTV-resistant citrus [11]. This strategy was also applied to target CTV vector *Citrus psylla* to prevent CTV spread. For example, when the genes related to wing development were knocked out in *C. psylla*, its descendants were intrinsically deficient of wings, thus limiting the spread of CTV [12].

As a counter-defense strategy, many viruses evolve viral suppressor of RNA silencing (VSR) to inhibit siRNA-based host defense [13,14]. It was reported that the virulent CTV T36 genome encodes three VSRs with p20 and p23 suppressing intracellular silencing and p20 and CP suppressing intercellular silencing signals [15]. It is worth noting that p23 has the highest expression and strongest suppression activity among these three VSRs [16]. Interestingly, siRNA deep-sequencing analysis showed that the antiviral RNA silencing mechanism primarily targets the 3’ end of the CTV genome, specifically the p23 region [17]. VSR activity largely determines the virulence of a virus in its host [18]. Currently, studies on CTV VSRs mostly focus on virulent strains. Their homologues from weak strains are not well characterized. Here, we cloned and sequenced a weak strain N4 found in China, and BLAST analysis revealed that N4 is closely related to the mild T30 strain. Local silencing suppression activity was confirmed for p20 and p23 of N4 by transient assay. Subcellular localization assay showed that N4p23 is localized in both nucleus and plasmodesmata. Bioinformatic analysis of p23 from available CTV sequences revealed conserved amino acids across different CTV strains. We conducted mutagenesis analysis on those conserved residues and identified some key residues with key roles in p23 suppressor activity and protein stability.

## 2. Materials and Methods

### 2.1. Virus Source and Plant Materials

Wild-type *Nicotiana benthamiana* (*N. benthamiana*) and transgenic N. benthamiana (16c) with a GFP gene were used for infection test as described previously [15]. Citrus leaves infected with CTV-N4 strain were provided by Professor Ni Hong from Huazhong Agricultural University. The source of the host of the N4 strain was sweet orange. CTV-N4 was inoculated onto Mexican lime, sweet orange, and Duncan grapefruit seedlings, but produced no visible symptoms on any of the biological indicators [19].

### 2.2. Cloning of N4 Sequence

Total RNA was extracted from the leaves of CTV-N4 infected citrus and the T30 sequence was used as a homologous reference sequence. The CTV was divided into 8 fragments (fragments 1 to 8) and reverse transcription primers were designed (CTV-fragment1R–8R) (Appendix A). After reverse transcription (M-MLV), PCR amplification (PrimeSTAR) was carried out with CTV-specific primers, and the positions of the fragments on the CTV genome are listed in Appendix A. Full-length CTV genomic RNA sequence was assembled from these fragments and submitted to the National Center for Biotechnology Information (NCBI) (MK779711).

### 2.3. CTV-N4VSR Expression Vector Construction and Transient Assays

The CTV-N4CP, CTV-N4p20, and CTV-N4p23 coding sequences were amplified and cloned into plant expression vector pH7LIC3.1.1 by homologous recombination, which expresses target protein with N-terminal 3*FLAG tags (Appendix A). The vectors pH7LIC3.0 (35S-EV), pH7LIC3.1.1-N4CP (35S-N4CP), pH7LIC3.1.1-N4p20 (35S-N4p20), pH7LIC3.1.1-N4p23 (35S-N4p23), and pH7LIC3.1.1-p19 (35S-p19) (Appendix A) were transformed into Agrobacterium GV3101. The GV3101 strains containing different vectors were then separately co-infiltrated with green fluorescent protein (GFP) expression vector pMS4 into the *N. benthamiana* leaves. The function of VSRs was examined by photographing under UV four days after infiltration (DAI). The optimized optical density (OD) value for infiltration was 0.2 for pMS4 Agrobacterium strain and 0.5 for the other strains.

### 2.4. Agroinfiltration, Subcellular Localization, and Live Cell Imaging

The CMV-C2b, CTV-N4p23, and CTV-T36p23 cDNAs were amplified and cloned by homologous recombination into vector pH7LIC5.1.1, which produces N-terminal GFP tagged proteins (Appendix A). The vectors pH7LIC6.0 (35S-GFP-EV), pH7LIC5.1.1-N4p23 (35S-GFP-N4p23), pH7LIC5.1.1-T36p23 (35S-GFP-T36p23), and pH7LIC5.1.1-C2b (35S-GFP-C2b) (Appendix A) were transformed into GV3101 strains. For transient expression, these Agrobacteria were infiltrated into *N. benthamiana* leaves individually at a concentration of 0.1 OD. The leaves were placed in the nucleus staining solution for 20 min before microscopic observation. Plasmodesmata (PD) staining solution was infiltrated into leaves 30 min before microscopic observation, and the leaves were photographed using a laser scanning confocal microscope (water objective) at 2 DAI. An excitation wavelength of 405 nm was used for observation of GFP florescence, 4′,6-diamidino-2-phenylindole (DAPI), and PD staining. PD staining solution was prepared as follows: solution A (containing 0.1% Aniline Blue in ddH_2_O) and solution B (containing1 M glycine, pH 9.5) were premixed at a ratio of 2:3 (*v/v*) one day before use. Nucleus staining solution contained 1 mM DAPI.

### 2.5. Amino Acid Sequence Alignment, Classification, and Phylogenetic Tree Analysis

In all, 250 p23 amino acid sequences were downloaded from NCBI (Appendix A) and classified into 20 groups (Appendix A) using CD-HIT with a parameter of 0.93-c. The phylogenetic tree of p23 sequences was generated using the ClustalW alignment of MEGA7.0 without truncation (Appendix A). Representative sequences were extracted from every group for sequence alignment to identify and present conserved residues (Appendix A).

### 2.6. Construction and Transient Expression of CTV-N4p23 Mutants with Point Mutation

Point mutation was introduced into N4p23 cDNA using the fusion PCR method with primers bearing desired point mutation. In total, 18 mutant N4p23 coding sequences (M1–18) were amplified and cloned into vector pH7LIC3.1.1 (35S-CCDB) as described earlier (Appendix A). The constructed vectors pH7LIC3.1.1-N4p23 (35S-p23-N4), pH7LIC3.0 (35S-EV), and PH7LIC3.1.1-p23M1–M18 (Appendix A) were transformed into Agrobacterium GV3101. The resultant Agro strains were co-infiltrated with pMS4 (35S-GFP) into *N. benthamiana* leaves. The function of VSRs was examined by photographing under UV of 365 nm wavelength at 5 DAI. The infiltrated leaves were photographed against UV at 4 DAI to compare the fluorescence brightness. The optimized OD value for infiltration was 0.2 for pMS4 and 0.5 for the other strains.

### 2.7. Western Blotting

Total proteins from samples (100 mg) were extracted with 300 μL 2× SDS buffer at 100 °C for 10 min and then transferred to ice. After centrifugation, 20 μL of supernatant was separated by electrophoresis at 80 v for 20 min, then at 120 v for 1 h in SDS-polyacrylamide gel (upper-layer gel 5%, lower-layer gel 10%). After electrophoresis, protein was transferred onto polyvinylidene fluoride (PVDF) membrane for 30 min using a semidry blotting method, then the membrane was incubated in 40 mL 1× Tris-buffered saline (TBS) with skim milk (0.5%) for 2 h, shaking at 100 r/s. Primary antibody was added into the incubation buffer and shaking continued for additional 2 h, then the membrane was washed with TBS 3 times for 10 min each time. Subsequently, the membrane was incubated with the secondary antibody for 1 h, shaking at 100 r/s, followed by 5 washes with TBS (10 min each time). The photographic developer was sprayed (A solution + B solution mix at 1:1) onto the membrane. The primary antibody was FLAG, GFP, Actin mouse anti-antibody, and the secondary antibody was goat anti-mouse (antibody concentration 1:10,000). Before reprobing with another antibody, the membrane was washed with absolute ethanol, incubated with 30% H_2_O_2_ at 37 °C for 15 min, and then washed with TBS three times (10 min each time).

### 2.8. Northern Blotting

RNA was extracted from *N. benthamiana* leaves with Trizol. Twenty μg RNA was separated by electrophoresis on denatured polyacrylamide gel (upper layer 6%, lower layer 15%) for 150 min at 300 v, then transferred to N+ membrane (GE Healthcare, Buckinghamshire, UK) for 90 min. Three μL of probe (10 mM GFP oligo mixture or miR156 oligo; Appendix A), 2 μL ddH_2_O, 1 μL 10× T4 PNK buffer, and 1 μL T4 PNK enzyme (NEB) were successively mixed on ice, then 3 μL ᵞ-^32^ P-ATP (PerkinElmer, Boston, USA) was added and the mixture was inoculated at 37 °C for 30 min. The labeled probe was added to the hybridization solution (Sigma, Spruce Street, Saint Louis, USA) with blotted membrane at 37 °C for 16 h. The N+ membrane was first washed by 2× SSC/0.1% SDS at 37 °C for 30 min, followed by washing with 0.5× SSC/0.1% SDS at 37 °C for 30 min, then the N+ membrane was exposed to the phosphor screen for 12 h. The screen was scanned with a phosphor screen scanner. Before reprobing with another probe, the membrane was stripped by incubation in preheated stripping buffer at 68 °C for 30 min, and the step was repeated once with fresh stripping buffer.

## 3. Results

### 3.1. Whole-Genome Sequencing and Cloning of CTV-N4 

As the CTV sequence is too long to amplify as a full-length cDNA, eight pairs of primers were designed to amplify the N4 as eight fragments (Figure 1A). After RT-PCR, eight bands of 2981 bp, 1450 bp, 1777 bp, 3019 bp, 3099 bp, 3094 bp, 2724 bp, and 1667 bp were obtained (Figure 1B). All fragments were cloned into pEasy vector and sequenced by Sanger sequencing. Alignment of the N4 sequences with T30 sequences showed 99.6% identity, indicating low genetic variation from the N4 strain.

### 3.2. CTV-N4p23 is a Strong Silencing Suppressor

Since the silencing suppressors of CTV mild strains have been rarely studied, we set out to investigate the function of the CTV-N4 silencing suppressors. CTV-N4 CP, p20, and p23 were cloned into pH7LIC3.1.1 vector (Figure 2A) and each was co-expressed with 35S-GFP in 16c and *N. benthamiana* by agroinfiltration. Subsequently, their green fluorescence signals were examined under UV light. The results revealed that there were no differences in fluorescence signals in the leaf patch where 35S-GFP co-expressed with empty vector (EV) and CTV-N4CP (Figure 2B). The fluorescence signal was slightly higher in the patch co-expressing N4p20 and 35S-GFP than in the patch co-expressing EV and 35S-GFP (Figure 2C), while it was significantly stronger in the area co-expressing N4p23 and 35S-GFP than in the area co-expressing EV and 35S-GFP, which was similar to that of the tomato bushy stunt virus (TBSV) p19 (positive control), which is a potent silencing suppressor [20] (Figure 2D), indicating that N4p23 had the strongest local silencing suppression activity among the three CTV proteins.

### 3.3. Subcellular Localization of N4p23

To determine the subcellular localization of N4p23, the free GFP, nuclear localized GFP-C2b [21], GFP-N4p23, and GFP-T36p23 were expressed individually in *N. benthamiana* leaves by agroinfiltration. Microscopic observation revealed that free GFP was localized in various parts of *N. benthamiana* cells (Appendix A), while GFP-C2b was only localized in the nucleus (Appendix A). In both GFP-N4p23 and GFP-T36p23 expressing cells, green fluorescence was observed in the nucleus and cell membrane (Figure 3A,I). After the leaves were stained with DAPI, the nucleus was clearly marked by blue fluorescence under UV (Figure 3B,J). The green and blue fluorescence could overlap (Figure 3C,D,K,L). These data show that both p23-N4 and p23-T36 could be localized on the nucleus. Further analysis of p23 localization on cell membrane revealed discrete dots (Figure 3E,M). After the leaves were stained by PD staining solution, the PD was marked by blue fluorescence under UV (Figure 3F,N). The green and blue fluorescence partially overlapped (Figure 3G,H,O,P). These results suggest that both N4p23 and T36p23 are also localized on the PD. 

### 3.4. Identification of Conserved Amino Acids in CTV-p23

Considering the potent silencing suppressor function of p23 and its multifunctionality [22], we next attempted to identify the conserved residues in p23, which may be helpful to elucidate the mechanisms for its diverse functions. We conducted a similarity analysis of 250 amino acid sequences of p23 found in NCBI (Appendix A). After clustering the sequences with identities over 93%, a total of 20 groups were obtained. One sequence was selected from each group for phylogenetic analysis. The phylogenetic tree can be divided into five clusters. P23 sequences in the first cluster show higher sequence identity to RB, T30, and VT p23 sequences, among which the P60 is identical to our newly sequenced N4 p23. P23 sequences in the second cluster show higher sequence identity to VT and T3 p23 sequences. The third cluster contains p23 genes with higher identity to T36 sequence. There is only one p23 gene in the fourth and fifth clusters, which is similar to the T30 and T36 p23 sequences, respectively (Figure 4B). The analysis showed that the N terminus of p23 was less conserved than its middle and C-terminus and several highly conserved amino acid blocks were identified (Figure 4A). In order to identify conserved residues that may be important for p23 suppressor activity, we set out to generate p23 mutants with alanine substitutions for those conserved sites. In total, 18 conserved sites of 1–3 amino acids were chosen and point mutations were introduced into the N4p23 expression vector (Figure 4C).

### 3.5. Conserved Amino Acids of p23 were Important for Its Suppressor Activity

To test the VSR activity of the above p23 mutants, those mutants and wild-type N4p23 were co-expressed with 35S-GFP in *N. benthamiana*. After several repeats, we identified six mutants (M4, M5, M6, M14, M17, and M18) showing a consistent level of suppression of GFP silencing. The 35S-GFP co-expressed with M5, M6 and M14 showed clearly weaker fluorescence than that co-expressed with WT N4p23 (Figure 5A,B,D). However, 35S-GFP co-expressed with M4, M17 and M18 showed similar level of florescence as that co-expressed with WT N4p23 (Figure 5C,E,F). Western blot analysis of GFP and Actin protein level showed that co-expression of EV and 35S-GFP resulted in the lowest normalized GFP level due to induction of silencing against GFP, while co-expression with WT N4p23 enhanced GFP accumulation due to suppression of GFP silencing (Figure 5G,H). The normalized GFP protein levels in samples co-expressing M5, M6, and M14 were similar to those in samples co-expressing EV, but lower than that in sample co-expressing WT p23, while the normalized GFP levels in samples expressing M14, M17, and M18 were higher than that in sample co-expressing EV but similar to the level in sample co-expressing WT N4p23 (Figure 5H). These data suggest that suppressor activity of M5, M6, and M14 was significantly impaired by point mutations, while that of M4, M17, and M18 was only moderately affected.

To further explore the mechanisms by which the above mutation affected p23 suppressor activity, Western blot was conducted to analyze the p23 protein levels, and results showed that M5 and M6 proteins were barely detectable, M4 and M14 proteins were detected at significantly reduced levels compared to wild-type p23, and M17 and M18 proteins accumulated to similar levels as wild-type p23 (Figure 5G,I). GFP siRNA and endogenous miR156 were also detected by Northern blot and quantified. The level of GFP siRNA signal in each sample was normalized to miR156 and the sample expressing wild-type p23 accumulated the lowest level of GFP siRNA, while the negative control sample expressing EV accumulated the highest level (Figure 5G,I), suggesting that p23 suppresses silencing by efficiently reducing siRNA accumulation. M17- and M18-expressing leaves accumulated similar levels of GFP siRNA to wild-type p23-expressing leaves. Interestingly, GFP siRNA levels in M5-, M6-, M14-, and M4-expressing leaves were similar to that in EV-expressing control leaf (Figure 5I), though M4 retained nearly wild-type suppressor activity in terms of GFP accumulation levels.

### 3.6. Conserved Amino Acids in p23 Protein Played Differential Roles in Protein Stability and Suppressor Activity

It is reasonable to speculate that a suppressor must be stably present in the cell in order to suppress silencing. Because transient expression of p23 is also targeted by gene silencing, there will be positive feedback between protein stability and suppressor activity. Dramatic variation of p23 mutant protein levels detected in previous experiments raised the question that those point mutations in M5, M6, M4, and M14 may have a direct role in p23 suppressor activity or indirectly abolish suppressor activity by reducing p23 protein stability. To test the impact of these point mutations on p23 protein alone, they were co-expressed with EV or TBSV p19. As a control, wild-type p23 was co-expressed with p19 or EV. Western blot and normalization analysis showed that M5 and M6 remained undetectable in the presence of p19 expression, while wild-type p23 accumulated to similar levels in the presence or absence of p19 expression (Figure 6A), suggesting that point mutation in M5 and M6 abolished p23 protein accumulation. In contrast, M4 and M14 protein levels in the presence of p19 expression were three to seven times higher than those in the absence of p19 (Figure 6B), suggesting that point mutation in M4 and M14 does not abolish p23 protein stability when p19 suppresses silencing against p23.

## 4. Discussion

CTV is one of the most destructive viruses, causing severe losses in the citrus industry [1]. Mild strains of CTV can protect citrus trees from decline [23], indicating the significance of conducting research on these strains. VSRs are important components of viral infection. However, previous studies on the silencing suppressor function of CTV were mainly focused on the VSRs of severe CTV strains [15,24,25]. Considering the lack of literature on mild strains of CTV, we cloned and sequenced the N4 weak strain in China and compared it with the six groups of known CTV strains. It was found that N4 is closely related to T30. We then tested the suppressor activity of N4 CP, p20, and p23 by co-infiltration assay and found that p23 has the strongest and p20 has moderate local silencing suppressor activity while CP does not, which is similar to the situation of the severe strain T36 [15] (Figure 2). T36p23 was shown to possess a bipartite nucleolar localization signal (NoLS) encompassing its zinc finger motif from 50 to 86 amino acids and a basic region from 143 to 158 AA [24]. It also localizes to plasmodesmata determined by AA from 143 to 158 [24]. Our localization study showed that both GFP-T36p23 and GFP-N4p23 were localized to nucleus and plasmodesmata (Figure 3). The plasmodesmata localization is consistent with previous findings [24]. However, for both T36p23 and N4p23, we observed nuclear localization but not nucleolar localization (Figure 3), which is similar to the localization pattern of the NoLS mutant of T36p23 studied previously [24]. This discrepancy could be due to the different location of the GFP tag in the fusion protein. In our study, we fused GFP at the N-terminus, which is close to the long zinc finger motif of NoLS, thus the N-terminal GFP fusion may block the zinc finger region from interActing with nucleolar importing machinery, whereas in the previous study the GFP tag was fused to the C-terminus of T36p23, which may have less impact on nucleolar localization. Our results together with previously published results suggest that the position of GFP fusion may affect target protein subcellular localization, especially for studying protein nucleolar localization.

To further study the mechanism of p23 suppressor activity, we made point mutations at conserved p23 residues. Combining the suppressor assay, protein stability, and siRNA accumulation analyses, we can classify the mutants into three classes: those that do not significantly affect suppressor activity and protein stability, such as M17 and M18; those that abolish protein stability and thus reduce suppressor activity, such as M5 and M6; and those that only reduce suppressor activity, such as M4 and M14. M17 and M18 have mutations at P175 and V182, respectively, in N4p23. C-terminal truncation mutant, p23Δ158-209, of T36p23 was shown to enhance PVX virulence like wild-type p23, and transgenic citrus expressing this mutant also induced stem pitting like wild-type p23 [24], suggesting that C-terminal 51 AA are not required for wild-type suppressor activity, which is consistent with the minimum impact of mutation in M17 and M18 on N4p23 suppressor activity. M5 and M6 have mutations at E95V96 and M99L100, respectively. These mutations are located in highly conserved regions and involve substitution of hydrophobic residues by alanine (Figure 4). These mutations may have a deleterious effect on protein folding, thus rendering the protein unstable, and are barely detectable by Western blot analysis (Figure 5 and Figure 6). M4 and M14 have mutations at Q93 and R143E144, respectively. These two mutants showed moderate suppressor activity, which is correlated with their own protein levels (Figure 5G). Co-expression with p19 greatly enhanced their protein levels even higher than wild-type p23 levels, indicating that those mutations did not affect protein stability. In summary, our results show that p23 suppressor activity in transient assays depends on its protein level, which in turn depends on factors affecting both protein stability and suppressor activity. In addition, we identified key residues specific to p23 stability, E95V96 and M99L100, and to suppressor activity, Q93 and R143E144.

It is interesting to note that co-expressing M4 and 35S-GFP resulted in similar levels of GFP accumulation to co-expressing 35S-GFP and M17 (or M18), but GFP siRNA in M4-expressing leaf accumulated to levels as high as those in the negative control sample expressing EV (Figure 5G). It is possible that induction of silencing, a high level of siRNA accumulation, and repression of protein accumulation may occur in different stages and p23 may suppress RNA silencing at different stages depending on its suppressor activity. The wild-type p23 has full silencing suppressor activity and itself accumulates to high levels to suppress silencing at the induction stage, thus less GFP siRNA is produced and GFP protein accumulates to high levels. When p23 suppressor activity is moderately impaired in M4, p23 activity and concentration in the cell are not high enough to suppress induction of silencing, thus allowing amplification of siRNAs. However, the remaining M4 suppressor activity and concentration are still enough to inhibit the siRNA-mediated repression of GFP protein synthesis. Our study provides new insights on the p23 suppressor mechanism and a basis for further investigation of the interaction between plants and the mild CTV strain.

## Figures and Tables

**Figure 1 viruses-11-00782-f001:**
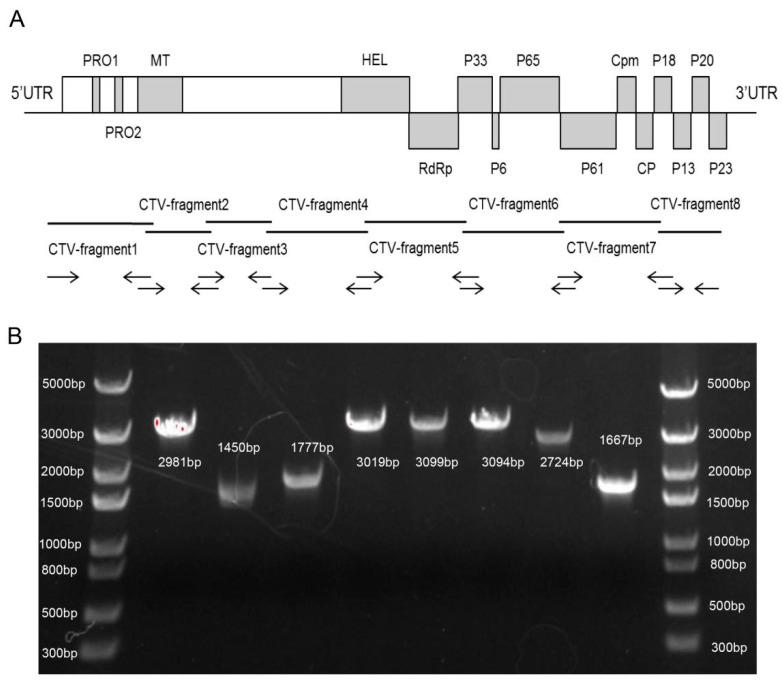
Citrus tristeza virus (CTV)-N4 cloning. (**A**) CTV genome organization and schematic representation of eight fragments amplified individually. (**B**) Product of each fragment cloned from CTV-N4. DNA bands from left to right are Trans5k DNA marker, successively purified CTV-N4 fragments 1 to 8, Trans5k DNA marker; and fragment length is indicated in the figure.

**Figure 2 viruses-11-00782-f002:**
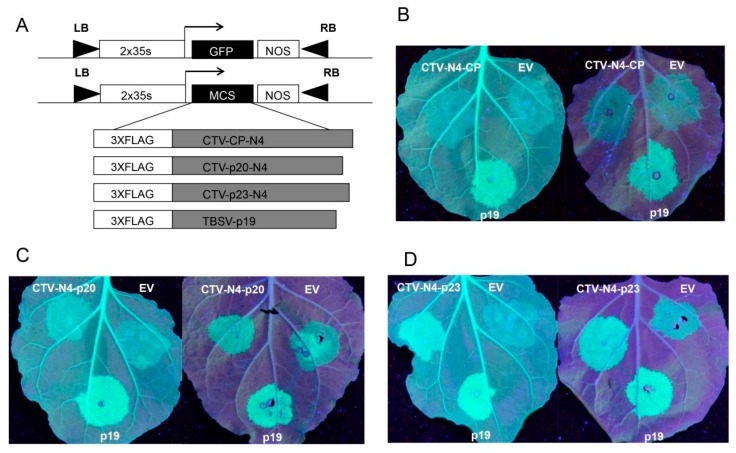
Silencing suppressor function of CTV-N4 proteins (CP) p20 and p23. (**A**) Schematic representation of expression cassettes for 35S-GFP, N4CP, N4p20, N4p23, and p19. The 2x 35S promoter, terminator (NOS), and N-terminal 3xFLAG tags are shown as open boxes. Filled triangles represent T-DNA left (LB) and right (RB) border. For B, C and D, the 16c leaf is on the left and the Nicotiana benthamiana wild type leaf is on the right. (**B**) Representative picture of leaves co-expressing 35S-GFP with N4CP, negative control empty vector (EV), and positive control p19, respectively. (**C**,**D**) N4p20 and N4p23 expressing leaves as in B.

**Figure 3 viruses-11-00782-f003:**
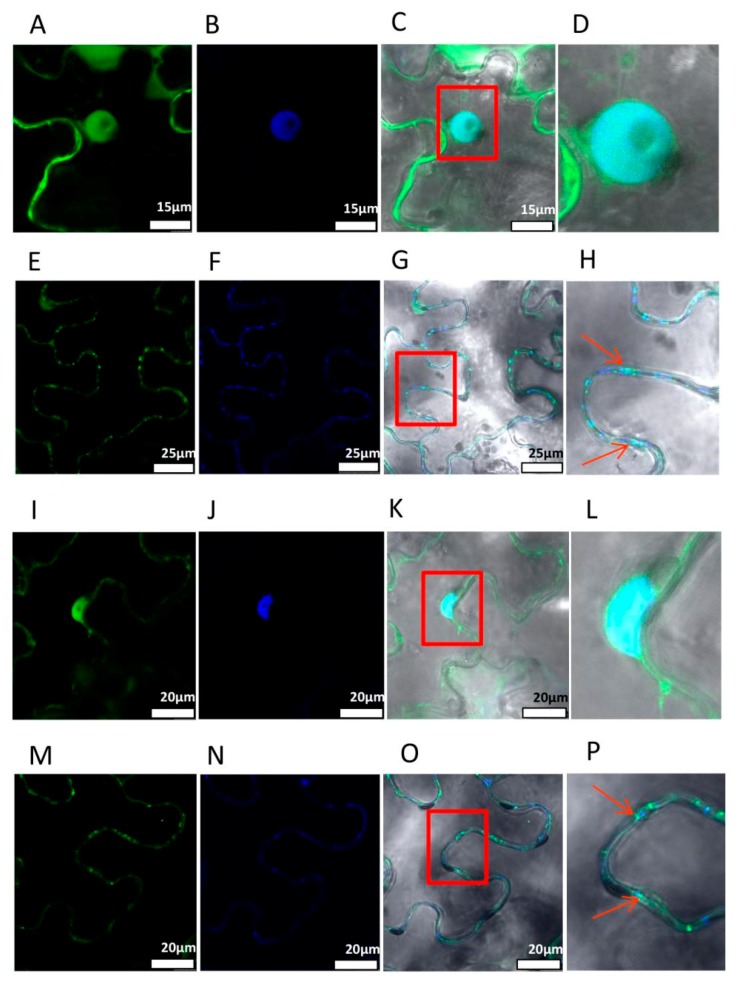
Subcellular localization of CTV-N4p23 and T36p23. (**A**) GFP-N4-p23 nuclear green fluorescence. (**B**) Nucleus excited in UV region and marked by blue fluorescence. (**C**) Overlap of two fluorescence signals. (**D**) Magnified view of overlap (red box in C). (**E**) GFP-N4p23 PD green fluorescence. (**F**) PD excited in the UV region and marked by blue fluorescence. (**G**) Overlap of two fluorescence signals. (**H**) Magnified view of overlap (red box in G); arrow indicates overlapping PD. (**I**) GFP-T36p23 nuclear green fluorescence. (**J**) Nucleus excited in UV region and marked by blue fluorescence. (**K**) Overlap of two fluorescence signals. (**L**) Magnified view of overlap (red box in K). (**M**) GFP-T36-p23 PD green fluorescence. (**N**) PD excited in UV region and marked by blue fluorescence. (**O**) Overlap of two fluorescence signals. (**P**) Magnified view of the overlap (red box in O); the arrow indicates overlapping PD.

**Figure 4 viruses-11-00782-f004:**
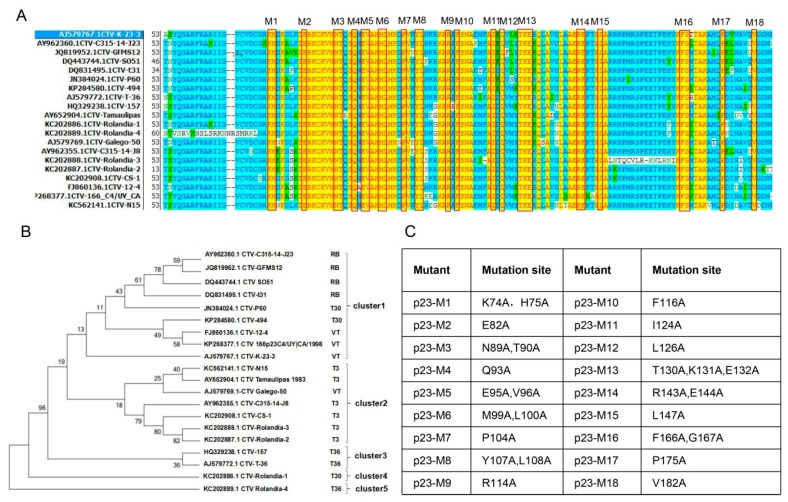
Identification of conserved residues in CTV-p23 for point mutation analysis. (**A**) Alignment of representative p23 sequences from each p23 sequence group. The 20 representative sequences of p23 selected by phylogenetic tree alignment were used for amino acid sequence alignment. Yellow region is conserved sites with identical sequences, selected for mutation and marked in the figure. (**B**) Gene phylogenetic tree analysis of p23 amino acid sequences after grouping. CTV genotypes of p23 sequence for comparison are marked in the figure. (**C**) List of p23 point mutation sites in each mutant p23 construct. All selected conserved sites of amino acids were mutated to alanine. Taking M1 as an example, 74K and 75H were mutated to A, and are thus noted as K74A, H75A.

**Figure 5 viruses-11-00782-f005:**
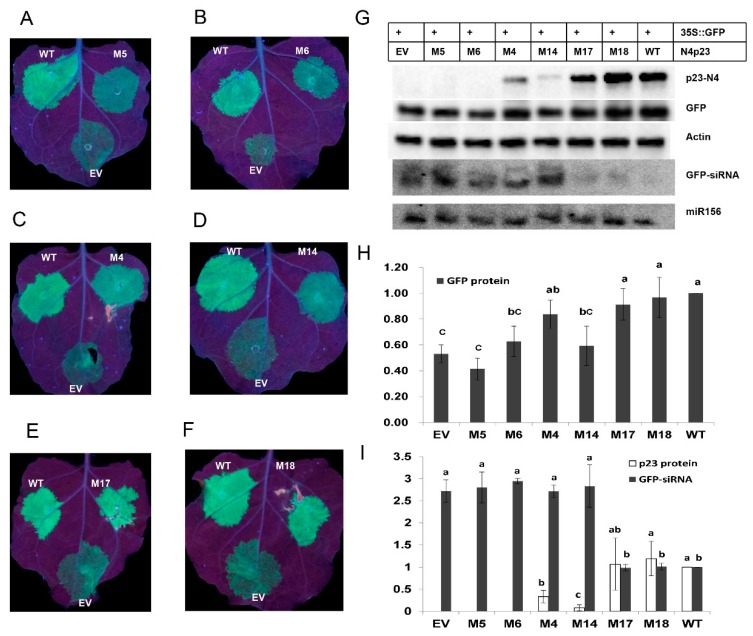
Characterization of p23 mutants in transient assays. (**A**–**F**) 35S-GFP was co-infiltrated with wild-type p23 (WT), negative control (EV), and indicated p23 mutant in three patches. Representative leaves were photographed under UV light at 5 days after infiltration (DAI). (**G**) Western and Northern blot analysis of co-infiltration samples. Top two rows indicate combination of co-expressed genes. Proteins (N4p23, GFP, and Actin) detected by Western blot are indicated to the left of each blot. Small RNAs (GFP and miR156 siRNA) detected by Northern blot are also indicated to the left of each blot. (**H**) Histogram of normalized expression value of GFP protein quantified from three independent experiments. (**I**) Histogram of normalized expression value of N4p23 protein and GFP siRNA quantified from western blots and northern blots in G. Columns with different letters indicate significant differences according to Duncan’s multiple tests (*p* < 0.05). Western blots and northern blots performed 3 and 2 independent experiments, respectively.

**Figure 6 viruses-11-00782-f006:**
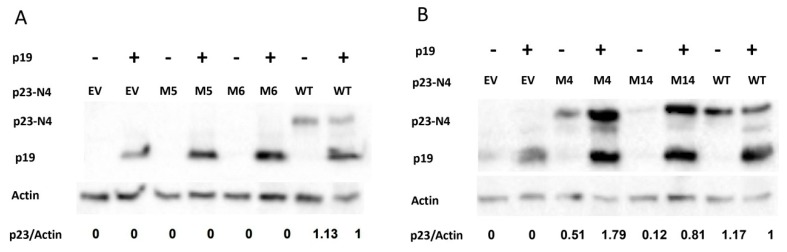
Point mutation in M5 and M6 but not M4 and M14 abolish p23 protein stability. p19 was co-infiltrated with N4p23 WT, M, or EV. Each row, top to bottom: p23 protein, p19 protein, and Actin protein and gray value ratio of p23 protein/Actin protein.

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
