# Peer review of "Identification of Key Residues Required for RNA Silencing Suppressor Activity of p23 Protein from a Mild Strain of Citrus Tristeza Virus"

_viruses, 2019, doi:10.3390/v11090782_

Round 1

Reviewer 1 Report

Overall, the work is well organized and written and scientifically sound presenting interesting and new data on VSRs in a mild strain of CTV.

Still, few minor changes are requested to improve the manuscript as follows:

At results section some figures need improvement. Figure 1 part B photo need to be improved since the figure's text described only the lane 1 to 8 and the first and last which I presume are the marker's lanes are forgotten. Describe also them and maybe write the size of the lanes on the right side for better reading. on the same part B of the figure change it with a version without the cut part on the marker (right side of photo) looks not good.

Again results section Figure 3 increase the character size on top of the scale bar is difficult to be read. Also, at part 0 change the colour of the character and give a small space between the scale bar and the measurement so can be seen better.

Figure 4 increase character size in the phylogenetic tree for better reading and furnish in the text some basic info about the tree realization.

Author Response

Comments and Suggestions for Authors

Overall, the work is well organized and written and scientifically sound presenting interesting and new data on VSRs in a mild strain of CTV.

Still, few minor changes are requested to improve the manuscript as follows:

Thank you for your suggestion, we have made the corresponding changes, the revised manuscript has been uploaded.All modifications are noted in the manuscript, and the modified picture is placed below the original image for easy comparison.

At results section some figures need improvement. Figure 1 part B photo need to be improved since the figure's text described only the lane 1 to 8 and the first and last which I presume are the marker's lanes are forgotten. Describe also them and maybe write the size of the lanes on the right side for better reading. on the same part B of the figure change it with a version without the cut part on the marker (right side of photo) looks not good.

Thank you for your suggestion, we have modified Figure 1B according to your suggestion, and the corresponding modifications in the manuscript are in lines 175-176.

Again results section Figure 3 increase the character size on top of the scale bar is difficult to be read. Also, at part 0 change the colour of the character and give a small space between the scale bar and the measurement so can be seen better.

Thank you for your suggestion, we have modified Figure 3 according to your suggestion.

Figure 4 increase character size in the phylogenetic tree for better reading and furnish in the text some basic info about the tree realization.

Thank you for your suggestion, we have modified Figure 4 according to your suggestion,and added the corresponding descriptions in the manuscript are in lines 232-237.

Reviewer 2 Report

The authors have went about to identify the suppressor protein in CTV and identify the key residues required for suppression in the  manuscript titled "Identification of Key Residues Required for RNA Silencing Suppressor Activity of p23 Protein from a Mild Strain of Citrus Tristeza Virus". My major concern is that the authors do not demonstrate silencing to demonstrate suppression property.  Transient expression of 35S-GFP in N. benthamiana is just an over-expression.  To demonstrate silencing the authors should be infiltrating 35S-GFP alone or along with candidate suppressor genes in N. benthamiana 16C plant (Developed in Prof. Baulcombe's lab), a trangenic plant expressing GFP at its threshold.  Hence this plant serves as the Rosetta stone to demonstrate silencing suppression.  Over-expressing GFP in N. benthamiana is not sufficient at all.  As the authors show in Fig 2 B-D you would see green fluorescence and relatively less amount in EV in Fig. 2B could be just because of less efficient infiltration.   If the authors argue the presence of GFP siRNA in empty vector lane of Fig. 5G, the problem is that there is no corresponding disappearance of GFP transcript.  Hence the authors should demonstrate silenicing suppression in N. benthamiana 16C.

Author Response

Comments and Suggestions for Authors

The authors have went about to identify the suppressor protein in CTV and identify the key residues required for suppression in the  manuscript titled "Identification of Key Residues Required for RNA Silencing Suppressor Activity of p23 Protein from a Mild Strain of Citrus Tristeza Virus". My major concern is that the authors do not demonstrate silencing to demonstrate suppression property.  Transient expression of 35S-GFP in N. benthamiana is just an over-expression.  To demonstrate silencing the authors should be infiltrating 35S-GFP alone or along with candidate suppressor genes in N. benthamiana 16C plant (Developed in Prof. Baulcombe's lab), a trangenic plant expressing GFP at its threshold.  Hence this plant serves as the Rosetta stone to demonstrate silencing suppression.  Over-expressing GFP in N. benthamiana is not sufficient at all.  As the authors show in Fig 2 B-D you would see green fluorescence and relatively less amount in EV in Fig. 2B could be just because of less efficient infiltration.   If the authors argue the presence of GFP siRNA in empty vector lane of Fig. 5G, the problem is that there is no corresponding disappearance of GFP transcript.  Hence the authors should demonstrate silenicing suppression in N. benthamiana 16C.

Thank you for your suggestion.we have made the corresponding changes, the revised manuscript has been uploaded.All modifications are noted in the manuscript, and the modified picture is placed below the original image for easy comparison.

We had conducted suppressor assays for CP, p20, p23 of CTV-N4 in both N. benthamiana 16c and N. benthamiana wild type plants. And in both cases, the suppressor activity for all three viral proteins was consistent. In the previous version of manuscript, we just described the results using the wild type plants for simplicity of description. Now we added back the image from 16c plants in Figure2, and we modified the legend in lines 196-197. The red ring around the EV plus 35S-GFP infiltrated patch is a clear indication of silencing signal movement from infiltrated patch to neighboring cells (Christophe Himber 2003 EMBO J). The reason the transient expression is transient is because induction of silencing during ectopic expression of the T-DNA encoded genes (Dunoyer P 2006 Nature Genetics). Increased expression of ectopic expression of T-DNA encoded GFP by co-expressing with candidate viral suppressor had been accepted as a standard method to test suppressor activity of RNA silencing in many publications including those recently published in Viruses.

1 Dey K. K., W. B. Borth, M. J. Melzer, M. L. Wang and J. S. Hu (2015). Analysis of pineapple mealybug wilt associated virus -1 and -2 for potential RNA silencing suppressors and pathogenicity factors. Viruses 7(3): 969-995.

2 Kenesi E., A. Carbonell, R. Lozsa, B. Vertessy and L. Lakatos (2017). A viral suppressor of RNA silencing inhibits ARGONAUTE 1 function by precluding target RNA binding to pre-assembled RISC. Nucleic Acids Res. 45(13): 7736-7750.

3 Martin-Hernandez A. M. and D. C. Baulcombe (2008). Tobacco rattle virus 16-kilodalton protein encodes a suppressor of RNA silencing that allows transient viral entry in meristems. J. Virol. 82(8): 4064-4071.

4 Nyiko T., F. Kerenyi, L. Szabadkai, A. H. Benkovics, P. Major, B. Sonkoly, Z. Merai, E. Barta, E. Niemiec, J. Kufel and D. Silhavy (2013). Plant nonsense-mediated mRNA decay is controlled by different autoregulatory circuits and can be induced by an EJC-like complex. Nucleic Acids Res. 41(13): 6715-6728.

5 Senshu H., Y. Yamaji, N. Minato, T. Shiraishi, K. Maejima, M. Hashimoto, C. Miura, Y. Neriya and S. Namba (2011). A dual strategy for the suppression of host antiviral silencing: two distinct suppressors for viral replication and viral movement encoded by potato virus M. J. Virol. 85(19): 10269-10278.

6 Thomas C. L., V. Leh, C. Lederer and A. J. Maule (2003). Turnip crinkle virus coat protein mediates suppression of RNA silencing in nicotiana benthamiana. Virology 306(1): 33-41.

7 Lakatos L., T. Csorba, V. Pantaleo, E. J. Chapman, J. C. Carrington, Y. P. Liu, V. V. Dolja, L. F. Calvino, J. J. Lopez-Moya and J. Burgyan (2006). Small RNA binding is a common strategy to suppress RNA silencing by several viral suppressors. EMBO J. 25(12): 2768-2780.

8 Mingot A., A. Valli, B. Rodamilans, D. S. Leon, D. C. Baulcombe, J. A. Garcia and J. J. Lopez-Moya (2016). The P1N-PISPO trans-Frame Gene of Sweet Potato Feathery Mottle Potyvirus Is Produced during Virus Infection and Functions as an RNA Silencing Suppressor. J. Virol. 90(7): 3543-3557.

Round 2

Reviewer 2 Report

Authors have addressed my concern